# Prognostic Impact of Infectious Agents After Definitive Treatment in Non-Small Cell Lung Cancer

**DOI:** 10.3390/cancers17203283

**Published:** 2025-10-10

**Authors:** Özlem Koca, Umur Kağan Kahya, Meltem Beydilli Şahiner, Rahmi Atıl Aksoy, Timur Koca, Aylin Fidan Korcum

**Affiliations:** 1Department of Medical Microbiology, Antalya Training and Research Hospital, University of Health Sciences, Antalya 07070, Turkey; 2Department of Radiation Oncology, Akdeniz University School of Medicine, Antalya 07070, Turkeymeltembeydilli@akdeniz.edu.tr (M.B.Ş.);; 3Department of Radiation Oncology, İzmir City Hospital, İzmir 35530, Turkey

**Keywords:** non-small cell lung cancer, chemoradiotherapy, infections, healthcare-associated infections, overall survival

## Abstract

**Simple Summary:**

Infections are common after definitive treatment in patients with non-small cell lung cancer (NSCLC), but their prognostic significance has not been clearly established. In this retrospective study, 214 patients with NSCLC who underwent definitive treatment were evaluated. Infections were found to be frequent within the first year after treatment, mainly caused by Gram-negative bacteria such as *Pseudomonas aeruginosa* and *Acinetobacter baumannii*. A substantial proportion of the isolated pathogens were healthcare-associated. Patients with positive cultures had significantly shorter overall survival. These findings demonstrate that infections are a strong prognostic factor in NSCLC after definitive treatment. Early microbiological evaluation, appropriate therapy, and strict infection prevention strategies are critical to improving overall survival in this vulnerable group.

**Abstract:**

Background: Infections are common complications in patients with non-small cell lung cancer (NSCLC) and may adversely influence clinical outcomes. Their prognostic impact after definitive treatment is not well established. This study aimed to investigate the incidence, microbiological profile, and prognostic significance of infections occurring within one year after definitive treatment in patients with NSCLC. Methods: We retrospectively analyzed patients with NSCLC who completed definitive treatment between 1 January 2016, and 31 December 2023. Microbiological culture results obtained within one-year post-treatment and inflammatory markers measured one month after treatment were evaluated. Pathogens were classified as healthcare-associated infection (HAI) or non-HAI agents. Overall survival (OS) was estimated using the Kaplan–Meier method, and prognostic factors were assessed using Cox regression analysis. Results: Among 214 eligible patients, 45 had positive microbiological cultures. Gram-negative bacteria predominated (n = 24), with *Pseudomonas aeruginosa* (n = 8) and *Acinetobacter baumannii* (n = 6) being the most frequently isolated species. Among all isolates, 20 Gram-negative and 6 Gram-positive microorganisms were identified as HAI pathogens. In multivariate analysis, culture positivity (HR: 2.75, *p* < 0.001) remained an independent prognostic factor for worse OS. Conclusion: Infections within the first year after definitive treatment, particularly those caused by HAI-related Gram-negative pathogens, are associated with reduced OS in NSCLC. Early microbiological diagnosis, targeted antimicrobial therapy, and strict infection prevention strategies may help improve outcomes in this high-risk population.

## 1. Introduction

Lung cancer remains the leading cause of cancer-related death in both men and women worldwide. Survival has improved in recent years due to advances in staging, earlier detection, and more effective treatments including radiotherapy, chemotherapy, immunotherapy and targeted therapies [1,2,3]. Patients with non-small cell lung cancer (NSCLC) are highly susceptible to infections. This is related to tumor-induced structural lung damage, treatment-related immunosuppression, and frequent invasive procedures [4,5]. Such infections complicate cancer therapy and increase both morbidity and mortality [6,7]. Infection-related deaths are a significant non-cancer cause of mortality, particularly among younger patients and those with advanced disease [8,9]. Despite these risks, there is limited evidence on the spectrum of infectious pathogens in NSCLC patients undergoing definitive treatment and on the prognostic impact of these infections. Slow and insensitive microbiological methods may delay effective treatment. In the absence of clear guidelines, empirical antibiotic use is common, which may promote antimicrobial resistance and worsen outcomes [10].

Advances in radiotherapy, such as intensity-modulated radiotherapy (IMRT) and volumetric modulated arc therapy (VMAT), have reduced toxicities like pneumonitis and esophagitis. However, treatment-related infections remain a persistent problem [11,12]. Chemotherapy-induced neutropenia and radiotherapy-induced lymphopenia further increase vulnerability to severe infections. Furthermore, targeted therapies such as new-generation EGFR tyrosine kinase inhibitors have also been reported to influence host immunity and contribute to infection risk [13].

Given the lack of definitive evidence, further research is needed to clarify the microbiological profile of post-treatment infections and their prognostic relevance in unresectable NSCLC. The present study aimed to identify infectious agents occurring within one year after definitive treatment and to evaluate their association with overall survival (OS) in patients with NSCLC.

## 2. Materials and Methods

### 2.1. Patient Selection

Between 1 January 2016, and 31 December 2023, 597 patients with NSCLC who received anti-cancer treatment were retrospectively evaluated. 383 of them were excluded from the study because they did not meet the inclusion criteria. A total of 214 patients who underwent definitive treatment were included in the final cohort. The study design flowchart is shown in Figure 1.

The inclusion criteria for this study were as follows: (1) histopathologically confirmed NSCLC deemed inoperable before treatment, with staging verified by positron emission tomography and computed tomography; (2) completion of definitive treatment, including concurrent chemoradiotherapy, induction chemotherapy followed by concurrent chemoradiotherapy, or definitive radiotherapy alone; and (3) no evidence of active infection at the beginning of follow-up, based on laboratory findings and microbiological records. Patients were excluded if they had undergone surgical resection, received palliative radiotherapy, or had a known diagnosis of interstitial lung disease prior to cancer diagnosis. Patients who received immune checkpoint inhibitors, maintenance immunotherapy, or molecularly targeted therapies during the study period were also excluded.

### 2.2. Treatment and Follow-Up

A total of 151 patients received concurrent chemoradiotherapy, 59 patients were treated with induction chemotherapy followed by chemoradiotherapy, and 4 patients received radiotherapy alone. All patients underwent external beam radiotherapy delivered via three-dimensional conformal radiotherapy, IMRT, or helical IMRT techniques. Radiotherapy was administered with a daily fraction dose of 2 Gy to a total dose ranging from 60 to 66 Gy. Following treatment completion, patients were monitored through routine physical examinations and radiological imaging as part of standard follow-up protocols.

### 2.3. Data Collection

Clinical data were collected retrospectively from the institutional electronic medical records. Sociodemographic and physiological variables were also recorded, including education level, place of residence, income level, body mass index (BMI), and the presence of comorbid conditions. Comorbidity burden was assessed using the Charlson comorbidity index (CCI) [14]. Laboratory parameters, including C-reactive protein (CRP), platelet count, lymphocyte count, neutrophil count, and monocyte count, measured at the first month following completion of definitive treatment were recorded. In addition, microbiological culture results from sputum, tracheal aspirate, and blood samples obtained within one year after completion of therapy were reviewed. In patients with positive culture results, the isolated microorganisms were identified and classified according to whether they met the definition of healthcare-associated infection (HAI). HAI was defined according to the Centers for Disease Control and Prevention (CDC) / National Healthcare Safety Network (NHSN) surveillance criteria, which classify cases as infections that were not present or were incubating at the time of hospital admission [15]. Bacterial identification was performed using matrix-assisted laser desorption–ionization time-of-flight mass spectrometry (MALDI-TOF MS) (Bruker Daltonics, Bremen, Germany). Antibiotic susceptibility tests were performed using the fully automated BD Phoenix System (Becton Dickinson, Franklin Lakes, NJ, USA) and evaluated according to the European Committee on Antimicrobial Susceptibility Testing (EUCAST) criteria. Antibiotics that were not included in the EUCAST-based institutional testing panel for the isolated infectious agents were marked as untested.

### 2.4. Statistical Analysis

All statistical analyses were performed using IBM SPSS, version 24.0 (IBM Corp., Armonk, NY, USA). A *p*-value of <0.05 was considered statistically significant. Descriptive statistics were used to summarize patient and microbiological characteristics. OS was defined as the time from the completion of definitive treatment to either the date of death or the date of last follow-up. Receiver operating characteristic (ROC) curve analysis was conducted to evaluate the predictive performance of variables for OS, with optimal cut-off values selected at the point where sensitivity and specificity were closest [16]. To identify potential prognostic factors for OS, univariate Cox regression analyses were performed. Variables that demonstrated statistical significance in univariate analysis were subsequently included in a multivariate Cox regression model to determine independent prognostic factors for OS. OS curves were estimated using the Kaplan–Meier method and differences between groups were compared using the log-rank test.

## 3. Results

Patient characteristics are summarized in Table 1. The median age was 65 years, and most patients were male (87.4%). The predominant histological subtype was squamous cell carcinoma (57.9%). Most patients had advanced-stage disease, with 85.6% presenting with stage III at diagnosis.

Among the study population, microbiological culture testing was performed in 111 patients (51.9%) during the one-year follow-up period following treatment. Of these, 45 patients had at least one positive culture result, while 66 patients had negative cultures. No culture samples were available for 103 patients. The most common sampling site was sputum (n = 29), followed by blood (n = 10) and tracheal aspirate (n = 6). Monomicrobial growth was observed in 38 cases, while 7 patients exhibited dual-organism growth. The distribution of sampling sites (sputum, tracheal aspirate, blood) and their HAI associations are shown in Table 2.

A total of 52 microorganisms were isolated. The distribution of pathogens by culture site is shown in Figure 2. Gram-negative bacteria were the most frequently detected pathogens, with *Pseudomonas aeruginosa* (*P. aeruginosa*) (n = 8) and *Acinetobacter baumannii* (*A. baumannii*) (n = 6) being the most common. Other isolated Gram-negative species included *Escherichia coli* (*E. coli*) (n = 3), *Stenotrophomonas maltophilia* (n = 2), *Klebsiella pneumoniae* (*K. pneumoniae*) (n = 1), *Acinetobacter pittii* (n = 1), *Haemophilus influenzae* (n = 1), *Moraxella catarrhalis* (n = 1), and *Salmonella* spp. (n = 1). Of the Gram-negative isolates (n = 24), 20 were classified as HAI pathogens. Gram-positive bacteria (n = 15) included *Staphylococcus aureus* (*S. aureus*) (n = 5), *Corynebacterium* spp. (n = 4), *Micrococcus luteus* (n = 2), *Enterococcus faecium* (*E. faecium*) (n = 1), and *Streptococcus* spp. (n = 3), and 6 of these were identified as HAI agents. Fungal isolates consisted primarily of *Candida albicans* (n = 11), as well as *Candida glabrata* (n = 1) and *Magnusiomyces capitatus* (n = 1).

Antibiotic resistance rates are shown in Table 3 and Table 4. Antibiotics marked with an asterisk (*) in Table 3 were not tested, as they were not included in the EUCAST-based institutional testing panel for the isolated infectious agents. The antibiotic resistance status of agents not listed in the table was also evaluated. Two *Stenotrophomonas maltophilia* strains were found to be susceptible to levofloxacin and intermediate to imipenem. *Salmonella* spp. was resistant to ampicillin and ciprofloxacin but susceptible to SXT and ceftriaxone. *Haemophilus influenzae* was resistant to AMC and susceptible to other antibiotics, including meropenem, ampicillin, cefotaxime, SXT, and cefuroxime axetil. *Moraxella catarrhalis* was resistant to erythromycin but susceptible to other antibiotics, such as AMC, cefotaxime, and levofloxacin. *Streptococcus pneumoniae* strains were resistant to oxacillin, SXT, levofloxacin, cefotaxime, and penicillin G but susceptible to vancomycin, erythromycin, meropenem, and daptomycin. No resistance was detected in *Streptococcus agalactiae* and *Streptococcus constellatus* strains, which were susceptible to penicillin G and other antibiotics. *Corynebacterium striatum* and *Corynebacterium amycolatum* strains were resistant to gentamicin, ciprofloxacin, tetracycline, rifampin, and clindamycin but susceptible to vancomycin and linezolid.

Among Gram-negative bacteria, the highest resistance was to ampicillin and AMC, while the most effective agents were imipenem, meropenem, and amikacin. Among Gram-positive bacteria, resistance was most common to ampicillin and penicillin G, while linezolid proved to be the most effective antibiotic. Candida glabrata strain was found to be susceptible to anidulafungin.

Optimal cut-off values were established through ROC curve analysis. The cut-off points for lymphocyte count, neutrophil count, and monocyte count were 1.015 × 10^3^/μL (sensitivity: 58.6%, specificity: 57.1%), 3.935 × 10^3^/μL (sensitivity: 44.8%, specificity: 44.9%), and 0.645 × 10^3^/μL (sensitivity: 52.6%, specificity: 52.0%), respectively. The optimal cut-off values for platelet count and CRP were 239 × 10^3^/μL (sensitivity: 39.7%, specificity: 39.8%) and 19.64 mg/L (sensitivity: 60.3%, specificity: 60.2%), respectively. Additionally, the optimal cut-off point for BMI was 24.75 kg/m^2^ (sensitivity: 49.1%, specificity: 49%). These values were subsequently used to categorize patients in survival analyses.

In univariate Cox regression analysis, diabetes mellitus (HR: 0.53, 95% CI: 0.34–0.82, *p* = 0.005), CCI (HR: 2.25, 95% CI: 1.50–3.36, *p* < 0.001), platelet count (HR: 1.59, 95% CI: 1.09–2.32, *p* = 0.01), lymphocyte count (HR: 0.63, 95% CI: 0.43–0.91, *p* = 0.01), CRP level (HR: 1.67, 95% CI: 1.15–2.42, *p* = 0.007), and culture positivity (HR: 3.29, 95% CI: 2.20–4.92, *p* < 0.001) were significantly associated with OS. In the multivariate Cox regression analysis, culture positivity (HR: 2.75, 95% CI: 1.78–4.27, *p* < 0.001) remained an independent prognostic factor for OS (Table 5).

Kaplan–Meier survival analysis revealed significantly reduced OS in patients with positive culture results compared to those with negative or no cultures (*p* < 0.001) (Figure 3). Similarly, patients with HAI-related pathogens demonstrated a trend toward worse OS compared to those without, with the difference approaching statistical significance (*p* = 0.08) (Figure 4). When stratified by the most common pathogens, *P. aeruginosa* and *A. baumannii* were each associated with worse OS (Figure 5).

## 4. Discussion

This study demonstrates that infections occurring within the first year after definitive treatment are associated with significantly reduced OS in patients with NSCLC. Culture positivity and the presence of HAI pathogens emerged as strong adverse prognostic indicators. These findings align with recent evidence underscoring that infections remain a major contributor to non-cancer mortality in NSCLC, even in the era of improved multimodal therapy [17,18].

The microbiological profile in this cohort was dominated by Gram-negative bacteria, most notably *P. aeruginosa* and *A. baumannii*, the majority of which were classified as HAI-related. Gram-positive isolates included methicillin-resistant *S. aureus* (MRSA), and fungal pathogens were predominantly *Candida albicans* with occasional *Candida glabrata*. This distribution is consistent with previous reports showing Gram-negative predominance in NSCLC patients treated with chemoradiotherapy or radiotherapy [19,20,21], as well as studies in elderly populations reporting frequent polymicrobial infections and opportunistic fungal involvement [22]. The predominance of HAI-related Gram-negative pathogens is particularly concerning, given their established association with prolonged hospitalizations, treatment delays, and worse oncologic outcomes.

Antimicrobial resistance patterns observed in this study have direct therapeutic implications. Gram-negative isolates exhibited high resistance to ampicillin and amoxicillin/clavulanic acid, while carbapenems and amikacin retained substantial efficacy. Among Gram-positive pathogens, resistance to penicillin and ampicillin was common, with linezolid demonstrating consistent activity. The isolation of multidrug-resistant *A. baumannii* and MRSA is in line with prior intensive care unit-based oncology series, which have emphasized the necessity of empiric regimens with broad coverage in high-risk patients [23]. The observed susceptibility of *Stenotrophomonas maltophilia* to levofloxacin, along with the preserved activity of glycopeptides and oxazolidinones against Corynebacterium species, highlights the importance of early microbiological identification to enable timely de-escalation of therapy, thereby minimizing toxicity and limiting further resistance development. Additionally, it should be noted that some antibiotics were not tested, as EUCAST-based institutional laboratory protocols recommend testing to agents considered clinically relevant for the isolated microorganisms.

Both treatment-related and disease-related mechanisms drive the adverse impact of infection on OS after definitive treatment in NSCLC. Treatment causes profound and sustained immunosuppression through neutropenia, mucositis, and impaired mucociliary clearance [5]. Radiotherapy has been shown to cause lymphopenia, which compromises the host’s immune defense [24]. Even at low doses, it can alter blood counts and other immune parameters, thereby increasing susceptibility to infections [25]. Qin et al. identified multiple chemotherapy cycles, radiotherapy, and elevated baseline neutrophil counts as independent predictors of pulmonary infection [19]. Tumor-related airway obstruction also predisposes patients to post-obstructive pneumonia, which is often difficult to eradicate and prone to recurrence [26]. Furthermore, tumor progression has been associated with reduced numbers and impaired function of natural killer (NK) cells, which weakens innate immune defense and may increase susceptibility to infections [27]. These risk factors were also present in our cohort, where HAIs and a predominance of Gram-negative pathogens were frequently observed.

Survival analyses in this study revealed a marked reduction in OS among patients with positive cultures, with a clear adverse trend among those infected with HAI-related pathogens. In the multivariate model, culture positivity remained an independent prognostic factor for worse OS. Kaplan–Meier survival curves and the clinical interpretation uniformly indicate that infection was detrimental to OS. These findings are consistent with the observations of Patel et al., who reported that pneumonia in NSCLC patients significantly increased both in-hospital mortality and overall comorbidity burden [18]. Similarly, Sarihan et al. found that nearly half of NSCLC patients developed infections during thoracic radiotherapy, most caused by Gram-negative bacteria, and that the presence of infection was associated with significantly shorter OS [4].

Although immune checkpoint inhibitor (ICI)-treated patients were excluded from this analysis, modern treatment paradigms frequently incorporate ICIs into first-line therapy. Evidence indicates that the combination of ICIs with chemotherapy increases infection risk through overlapping toxicities, such as ICI-related pneumonitis and chemotherapy-induced neutropenia [28,29,30]. By focusing exclusively on patients treated with radiotherapy and/or chemotherapy, our study provides a clearer assessment of the infection burden attributable to standard multimodality treatment, but it also suggests that infection rates could be even higher in current practice with ICI-based combinations.

The present findings underscore the need for vigilant post-treatment monitoring, rapid microbiological work-up, and early targeted antimicrobial therapy in patients with suspected infection. Strict adherence to infection prevention measures, including minimizing the use of invasive devices and reinforcing hospital infection control protocols, is crucial for reducing the incidence of HAI-related pathogens. Local antimicrobial resistance data should inform empiric treatment choices, with prompt de-escalation once susceptibility results are available. Attention should also be given to optimizing supportive care in patients with tumor-related airway obstruction, which may involve bronchial hygiene interventions or palliative procedures to restore airflow and reduce the risk of infection. Pneumococcal vaccination should be encouraged where appropriate, although routine antibiotic prophylaxis is not recommended due to the potential for resistance development and adverse effects [18,22].

This study has several limitations. Its retrospective design introduces the potential for selection bias, and the absence of culture results in nearly half of the cohort limits the ability to fully characterize the epidemiology of infection. Microbiological sampling was more likely to have been performed in clinically unstable patients, which may have inflated the association between culture positivity and mortality. Furthermore, treating infection as a fixed covariate in OS analysis does not account for its time-dependent nature and may underestimate its dynamic impact on survival. Future prospective, multicenter studies incorporating time-varying models and cause-specific mortality endpoints are warranted to validate these findings. The exclusion of ICI-treated patients may limit the applicability of this study to modern regimens; however, it also provides an opportunity to evaluate the independent effect of chemoradiotherapy-related infections on long-term outcomes.

## 5. Conclusions

Infections within the first year after definitive treatment, particularly those caused by HAI-related Gram-negative pathogens, are associated with a significant reduction in OS in NSCLC. Strengthening infection prevention strategies and implementing rapid, targeted antimicrobial treatment may mitigate this impact. These findings underscore the crucial importance of proactive infection surveillance and management as essential components of post-treatment care in this vulnerable population.

## Figures and Tables

**Figure 1 cancers-17-03283-f001:**
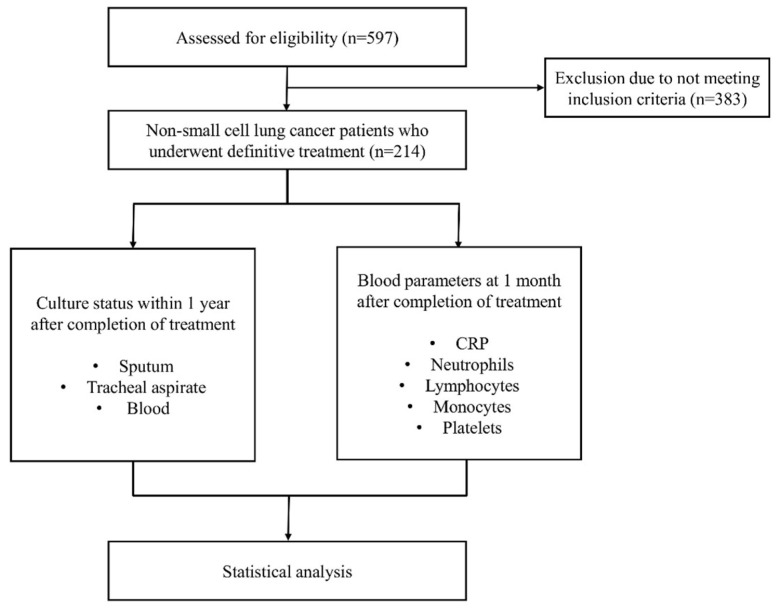
Study design flowchart.

**Figure 2 cancers-17-03283-f002:**
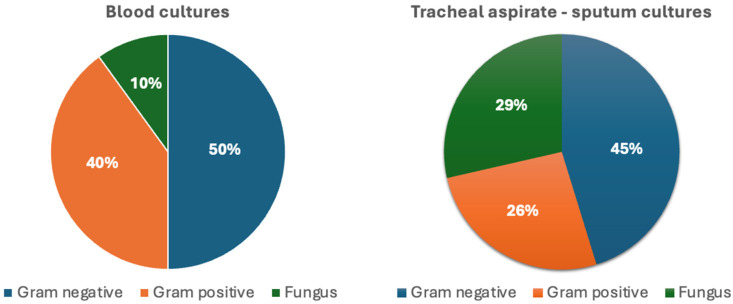
Distribution of pathogen groups isolated from blood and tracheal aspirate–sputum cultures.

**Figure 3 cancers-17-03283-f003:**
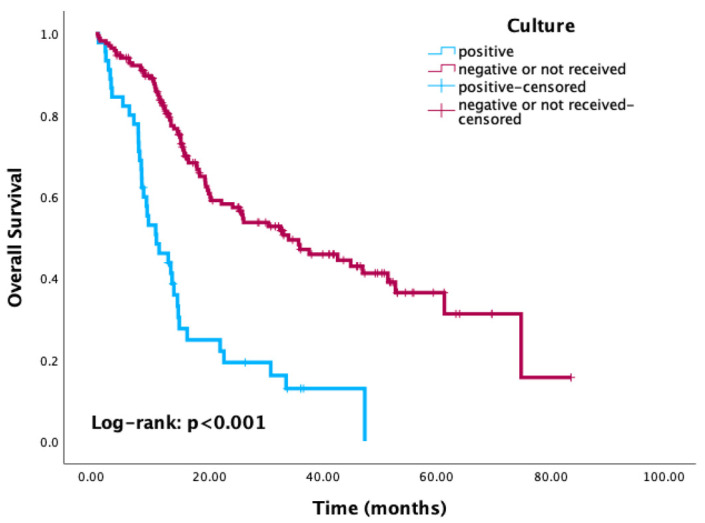
Kaplan–Meier curves of overall survival according to culture status.

**Figure 4 cancers-17-03283-f004:**
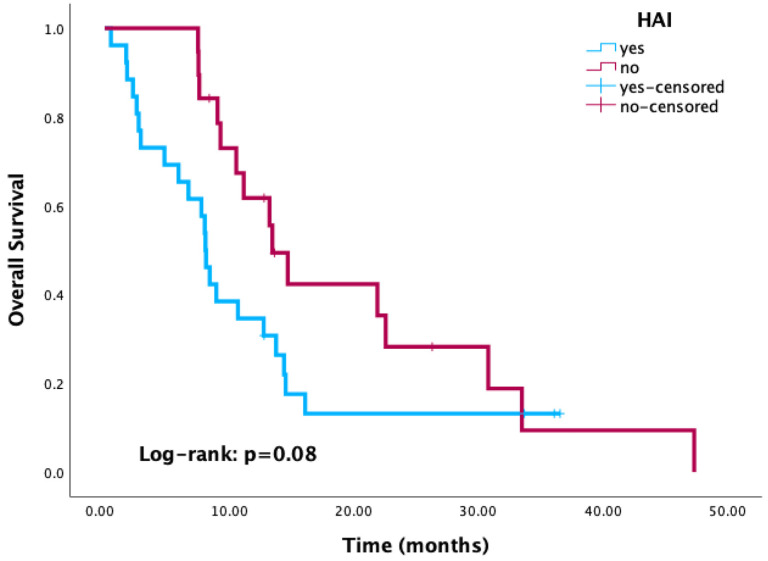
Kaplan–Meier curves of overall survival according to healthcare-associated infection status.

**Figure 5 cancers-17-03283-f005:**
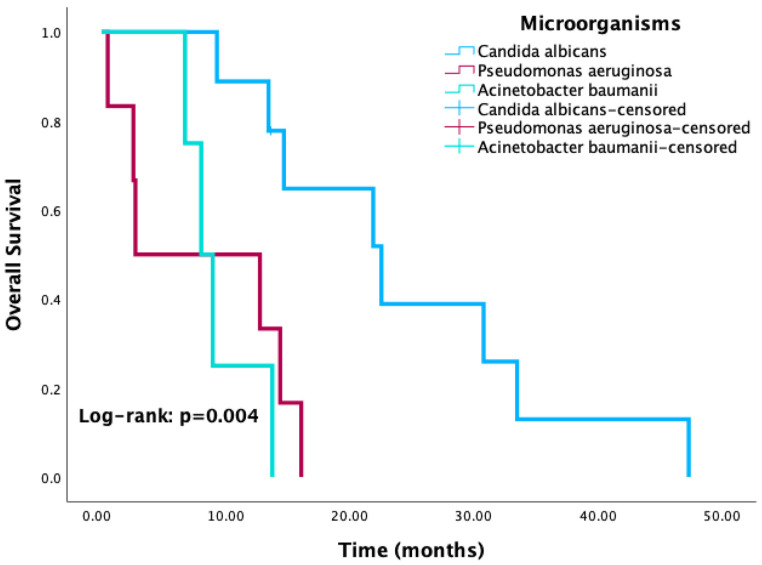
Kaplan–Meier curves for overall survival according to the three most common isolated microorganisms.

**Table 1 cancers-17-03283-t001:** Patient characteristics (n = 214).

Characteristic		n (%)
Gender	Male	187 (87.4%)
	Female	27 (12.6%)
Age	<65 years	103 (48.1%)
	≥65 years	111 (51.9%)
Education Level	Primary education	33 (15.4%)
	Lower secondary education	68 (31.8%)
	Upper secondary education	92 (43%)
	Tertiary education	21 (9.8%)
Place of Residence	Urban	157 (73.4%)
	Rural	57 (26.6%)
Income Level	Low	38 (17.7%)
	Middle	151 (70.6%)
	High	25 (11.7%)
BMI	<18.5 kg/m^2^	11 (5.1%)
	18.5–24.9 kg/m^2^	102 (47.7%)
	25.0–29.9 kg/m^2^	62 (29%)
	≥30 kg/m^2^	39 (18.2%)
Pulmonary Comorbidities	None	162 (75.7%)
	COPD	40 (18.7%)
	Asthma	9 (4.2%)
	Other	3 (1.4%)
Diabetes Mellitus	Yes	38 (17.8%)
	No	176 (82.2%)
Charlson Comorbidity Index	1–2 (mild)	11 (5.1%)
	3–5 (moderate)	152 (71.1%)
	≥6 (severe)	51 (23.8%)
Histopathology	Squamous cell carcinoma	124 (57.9%)
	Adenocarcinoma	61 (28.5%)
	Other	29 (13.6%)
T Stage	T1	15 (7%)
	T2	58 (27.1%)
	T3	48 (22.4%)
	T4	93 (43.5%)
N Stage	N0	42 (19.6%)
	N1	12 (5.6%)
	N2	140 (65.4%)
	N3	20 (9.4%)
TNM Stage	Stage I	5 (2.3%)
	Stage II	15 (7%)
	Stage III	183 (85.6%)
	Stage IV	11 (5.1%)
Treatment Modality	CRT	151 (70.6%)
	IC + CRT	59 (27.6%)
	RT alone	4 (1.8%)
RT Dose	60 Gy	169 (79%)
	66 Gy	41 (19.2%)
	Other	4 (1.8%)

BMI, body mass index; COPD, chronic obstructive pulmonary disease; CRT, chemoradiotherapy; IC, induction chemotherapy; RT, radiotherapy.

**Table 2 cancers-17-03283-t002:** Culture characteristics.

Characteristic		n (%) *
Culture Status	Positive	45 (21.1%)
	Negative	66 (30.8%)
	Not taken	103 (48.1%)
Culture Site	Sputum	29 (13.5%)
	Tracheal aspirate	6 (2.8%)
	Blood culture	10 (4.6%)
Growth	Monomicrobial growth	38 (17.7%)
	Dual-organism growth	7 (3.3%)
HAI	Yes	26 (12.2%)
	No	19 (8.9%)

HAI, healthcare-associated infection. * Percentages were calculated based on the total study population (n = 214).

**Table 3 cancers-17-03283-t003:** Gram-negative bacteria antibiotic resistance rates, n (%).

	*P. aeruginosa* (n = 8)		*A. baumannii* (n = 6)		*E. coli* (n = 3)		*K. pneumoniae* (n = 1)	
AK	-	-	6	(100)	-	-	-	-
AMC	*		*		*		1	(100)
Ceftazidime	8	(100)	*		-	-	-	-
Ciprofloxacin	*		*		3	(100)	-	-
Gentamicin	4	(50)	6	(100)	-	-	-	-
Imipenem	-	-	6	(100)	-	-	*	
Levofloxacin	5	(62.5)	6	(100)	3	(100)	*	
Meropenem	-	-	5	(83.3)	-	-	-	-
Piperacillin tazobactam	7	(87.5)	*		1	(33.3)	-	-
SXT	*		5	(83.3)	1	(33.3)	-	-
Cefepime	*		6	(100)	-	-	*	
Tigecycline	*		*		1	(33.3)	*	

AK, amikacin; AMC, amoxicillin/clavulanic acid; SXT, trimethoprim sulfamethoxazole. * Untested.

**Table 4 cancers-17-03283-t004:** *S. aureus* and *E. faecium* antibiotic resistance rates, n (%).

	*S. aureus* (n = 5)			*E. faecium* (n = 1)	
Cefoxitin	5	(100)			
Clindamycin	-	-			
Erythromycin	1	(20)	Ampicillin	1	(100)
Linezolid	-	-	HLGR	1	(100)
Rifampin	-	-	Linezolid	-	
SXT	-	-	Teicoplanin	1	(100)
Tetracycline	-	-	Vancomycin	1	(100)

SXT, trimethoprim sulfamethoxazole; HLGR, high-level gentamicin resistance.

**Table 5 cancers-17-03283-t005:** Univariate and multivariate Cox regression analysis for the prediction of overall survival.

		Univariate Analysis		Multivariate Analysis	
Variables	Cut-Off	HR (95% Cl)	*p*	HR (95% Cl)	*p*
Age (years)	<65 vs. ≥65	1.25 (0.86–1.81)	0.23		
Gender	Female vs. Male	1.01 (0.57–1.76)	0.97		
BMI (kg/m^2^)	<24.75 vs. ≥24.75	1.03 (0.71–1.50)	0.84		
Pulmonary comorbidity	Yes vs. No	0.94 (0.62–1.43)	0.80		
Diabetes mellitus	Yes vs. No	0.53 (0.34–0.82)	0.005	0.65 (0.40–1.07)	0.09
CCI	<6 vs. ≥6	2.25 (1.50–3.36)	<0.001	1.40 (0.87–2.25)	0.16
Histopathology	SCC vs. Other	0.99 (0.68–1.43)	0.97		
T stage	T1–T2 vs. T3–T4	1.23 (0.83–1.82)	0.29		
N stage	N0 vs. N1–3	1.51 (0.92–2.48)	0.09		
Treatment *	Induction vs. Other	1.10 (0.72–1.68)	0.64		
Lymphocyte, 10^3^	<1.015 vs. ≥1.015	0.63 (0.43–0.91)	0.01	0.81 (0.54–1.22)	0.32
Neutrophil, 10^3^	<3.935 vs. ≥3.935	1.32 (0.91–1.91)	0.14		
Monocyte, 10^3^	<0.645 vs. ≥0.645	0.83 (0.57–1.20)	0.33		
Platelet, 10^3^	<239 vs. ≥239	1.59 (1.09–2.32)	0.01	1.39 (0.92–2.10)	0.11
CRP (mg/L)	<19.64 vs. ≥19.64	1.67 (1.15–2.42)	0.007	1.29 (0.87–1.94)	0.20
Culture	Other vs. Positive	3.29 (2.20–4.92)	<0.001	2.75 (1.78–4.27)	<0.001

HR, hazard ratio; CI, confidence interval; BMI, body mass index; CCI, Charlson comorbidity index; SCC, squamous cell carcinoma; CRP, C-reactive protein. * Treatment: Induction = induction chemotherapy followed by definitive chemoradiotherapy; Other = definitive chemoradiotherapy or definitive radiotherapy.

## Data Availability

The data that support the findings of this study are available from the corresponding author upon reasonable request.

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
