# Peer review of "Prognostic Impact of Infectious Agents After Definitive Treatment in Non-Small Cell Lung Cancer"

_cancers, 2025, doi:10.3390/cancers17203283_

Round 1

Reviewer 1 Report

Comments and Suggestions for Authors

The paper is potentially interesting
However, corrections are needed to make the paper interesting and suitable for publication
1. In the introductory part, it is written that new therapies are used, but it should be said that in NSCLC, therapy is based on the use of new-generation TKI inhibitors, which are given to those with EGFR mutations, so the following papers should be added: PMID: 30406002, PMID: 40167836
2. Then, it should be emphasized that this therapy also has an impact on reducing immunity and side effects: PMID: 36644136
3. In the test where the therapy is mentioned, it should be stated whether and in what percentage the patients received TKI therapy or Immunotherapy, and whether they received Pdl-1 immune system blockers
4. Table 2 is incomplete and it is necessary to add a column in which the percentage of the total number of patients examined will be the percentage of participation
5. Table 3 should be converted into a pie chart where not the number but the percentage of individual pathogens from the total number. Do this for the sputum and especially for the blood
6. Regarding survival and infection, the groups positive for infection who were treated with radiotherapy should be separately distinguished in relation to those treated without radiotherapy in order to see the contribution of radiotherapy to infection.
7. It is known that radiotherapy induces changes in the blood count and in other immune parameters, even at low doses of radiation, which has been shown and should be cited: PMID: 40051520
8. It is known that during tumor progression, depending on the stage of the disease, there is a decrease in NK cells, and this should also be written in the discussion as one of the reasons for the presence of infections in tumors PMID: 20215803,

Comments on the Quality of English Language

need corection

Author Response

Author Response to Reviewer 1

1 - We sincerely thank the reviewer for this valuable comment. We agree that it is important to emphasize the role of new-generation tyrosine kinase inhibitors (TKIs) in the treatment of NSCLC patients with EGFR mutations. We have revised the Introduction accordingly and incorporated the suggested references (PMID: 30406002, PMID: 40167836).

2 - We also appreciate the reviewer’s suggestion to highlight the immunological and toxicity-related aspects of these therapies. As recommended, we have added a sentence to the Introduction to reflect that TKIs may influence host immunity and contribute to side effects, supported by the reference PMID: 36644136.

3 - We thank the reviewer for this important point. As clarified in the Methods section, patients who received molecularly targeted therapies (such as EGFR or ALK tyrosine kinase inhibitors), immune checkpoint inhibitors, maintenance immunotherapy, or PD-L1 inhibitors during the study period were excluded from the analysis. Therefore, none of the patients included in our cohort had received TKI therapy or immunotherapy, and the percentage was 0%. We have now explicitly stated this in the revised manuscript to avoid any ambiguity.

4 - We thank the reviewer for this comment. In Table 2, all percentages were calculated based on the total study population (n = 214), as specified in the table footnote. We believe this approach ensures consistency across the presented data.

5 - We sincerely thank the reviewer for this constructive comment. In accordance with the suggestion, Table 3 has been removed and replaced with pie charts (now presented as Figure 2), which illustrate the percentage distribution of individual pathogens isolated from blood and tracheal aspirate–sputum cultures. We believe that this visual representation provides a clearer understanding of the pathogen profile.

6 - We thank the reviewer for this thoughtful comment. In our study cohort, all patients underwent definitive radiotherapy (either as part of concurrent chemoradiotherapy, induction chemotherapy followed by chemoradiotherapy, or radiotherapy alone). Therefore, it was not possible to establish a comparison between patients treated with and without radiotherapy in relation to infection status. We have clarified this point in the Methods section to avoid any potential misunderstanding.

7 - We thank the reviewer for this insightful comment. As suggested, we have added a sentence to the Discussion highlighting that radiotherapy can induce changes in blood counts and other immune parameters, even at low doses. The recommended reference (PMID: 40051520) has been cited accordingly.

8 - We thank the reviewer for this important observation. As suggested, we have added a statement to the Discussion noting that tumor progression is associated with a reduction in NK cell numbers and function, which may contribute to increased susceptibility to infections in cancer patients. The recommended reference (PMID: 20215803) has been cited accordingly.

9 - The manuscript has been carefully revised to enhance clarity, grammar, and overall readability.

Reviewer 2 Report

Comments and Suggestions for Authors

The manuscript is promising with clinically relevant findings. However, there are a few points that deserve attention:

Methodology: 2.1. Patient Selection:

Study Period: Section 2.1, Paragraph 1 states: "Between January 1, 2016, and December 31, 2024." Considering that the journal's publication year is "Cancers 2025," the end of data collection in "December 2024" is unusual for a manuscript submitted for peer review. Please verify if the end date is correct.

Results: Untested Antibiotics: The use of an asterisk (*) for "untested" in Table 4 is clear. However, it would be beneficial to add a small note in the results or discussion section explaining, if possible, why some antibiotics were not tested.

Comments on the Quality of English Language

No comments

Author Response

Author Response to Reviewer 2

1 – We sincerely thank the reviewer for this thoughtful observation. You are correct that the year 2024 was mistakenly written in both the Abstract and the Methods sections. The correct study period is from January 1, 2016, to December 31, 2023. We have revised the text accordingly in both sections. We apologize for this oversight and greatly appreciate the reviewer’s careful reading, which has helped us improve the clarity and accuracy of our manuscript.

2 - We sincerely thank the reviewer for this valuable suggestion. As recommended, we have now added clarifications in the Methods, Results, and Discussion sections. Antibiotics marked with an asterisk in Table 4 were not tested because, according to institutional laboratory protocols based on the European Committee on Antimicrobial Susceptibility Testing (EUCAST) criteria, susceptibility testing is restricted to agents considered clinically relevant for the isolated infectious agents. We believe that these revisions enhance the clarity and methodological transparency of the manuscript.

3 - The manuscript has been carefully revised to enhance clarity, grammar, and overall readability.

Reviewer 3 Report

Comments and Suggestions for Authors

Dear Author,

I congratulate you on your work entitled “Prognostic Impact of Infectious Agents after Definitive Treatment in Non-Small Cell Lung Cancer.” In this manuscript, the authors preliminarily screened 214 NSCLC patients out of 597 individuals and analyzed blood inflammatory parameters along with bacterial culture reports. An interesting observation was that infections occurring within the first year after definitive treatment—particularly those caused by Gram-negative pathogens—were associated with reduced overall survival in NSCLC. This is a very complete and comprehensive study; however, I recommend some minor revisions to further strengthen the manuscript and broaden its impact.

Major Comments:

  1. Since most of the collected data focuses on post-therapeutic outcomes and survivability, it would be highly valuable to incorporate socio-financial and physiological variables from these patients. Examples may include: level of education, urbanicity, income, type of surgical procedure, post-surgery treatment/therapeutics, and nutritional status.

  2. I suggest including the Charlson Comorbidity Index to provide a more standardized assessment of patient comorbidities.

  3. Please consider reporting whether patients had any prior pulmonary disease history or comorbid conditions such as diabetes mellitus, as these may significantly influence outcomes.

These additions would enrich the clinical context of the findings and make the manuscript even more relevant for the field.

Comments on the Quality of English Language

Language is good

Author Response

Author Response to Reviewer 3

1 - We sincerely thank the reviewer for this valuable suggestion. In line with the comment, we have enriched our dataset by including additional socio-demographic and physiological variables. Specifically, patients’ education level, place of residence (urban vs. rural), income level, body mass index (BMI), and the presence of comorbid conditions such as COPD, asthma, and diabetes mellitus were collected and reported in Table 1. As no surgical intervention was performed in our cohort, surgical characteristics were not applicable. The Methods section has been revised accordingly.

2 - We sincerely thank the reviewer for this valuable suggestion. In accordance with the comment, we assessed comorbidity burden using the Charlson Comorbidity Index (CCI) and reported it in Table 1 as three categories: mild (CCI 1–2), moderate (CCI 3–5), and severe (CCI ≥6). We also incorporated CCI into our survival analyses. In univariate Cox regression, higher CCI was significantly associated with worse overall survival (HR: 2.25, 95% CI: 1.50–3.36, p<0.001). After adjustment for covariates in the multivariate Cox model, the association was attenuated and did not remain statistically significant (HR: 1.40, 95% CI: 0.87–2.25, p=0.16), whereas culture positivity remained an independent adverse prognostic factor. These additions have been reflected in the Methods, Results (Table 1 and Table 5), and the text accordingly.

3 - Thank you for the insightful comment. Interstitial lung disease was excluded by study design. We extracted COPD, asthma, and diabetes mellitus from physician-documented records and coded each as yes/no. These variables are presented in Table 1 as part of the descriptive analysis. We also evaluated their association with outcomes in the survival analysis (univariate and multivariate Cox models). The manuscript has been updated accordingly.

4 - The manuscript has been carefully revised to enhance clarity, grammar, and overall readability.

Round 2

Reviewer 1 Report

Comments and Suggestions for Authors

accept revision version

Reviewer 3 Report

Comments and Suggestions for Authors

The manuscript (MS) is presently in good shape for publication.